# The Role of Anti-Angiogenics in Pre-Treated Metastatic *BRAF*-Mutant Colorectal Cancer: A Pooled Analysis

**DOI:** 10.3390/cancers12041022

**Published:** 2020-04-21

**Authors:** Fabio Gelsomino, Andrea Casadei-Gardini, Daniele Rossini, Alessandra Boccaccino, Gianluca Masi, Chiara Cremolini, Andrea Spallanzani, Massimo Giuseppe Viola, Ingrid Garajovà, Massimiliano Salati, Maria Teresa Elia, Francesco Caputo, Chiara Santini, Alfredo Falcone, Stefano Cascinu, Emiliano Tamburini

**Affiliations:** 1Department of Oncology and Hematology, Division of Oncology, University of Modena and Reggio Emilia, 4121 Modena, Italy; fabiogelsomino83@yahoo.it (F.G.); andrea.spallanzani@gmail.com (A.S.); maxsalati@live.it (M.S.); francesco1990.caputo@libero.it (F.C.); santinic_90@libero.it (C.S.); 2Department of Translational Research and New Technologies in Medicine, Unit of Medical Oncology, Azienda Ospedaliero-Universitaria Pisana, University of Pisa, 56121 Pisa, Italy; danielerossini87@gmail.com (D.R.); boccaccina@gmail.com (A.B.); gl.masi@tin.it (G.M.); chiaracremolini@gmail.com (C.C.); alfredo.falcone@med.unipi.it (A.F.); 3Department of Surgery, Card. G. Panico Hospital of Tricase, 73039 Tricase, Italy; masgiuviola@yahoo.it; 4Medical Oncology Unit, University Hospital of Parma, 43121 Parma, Italy; ingegarajova@gmail.com; 5Department of Medical Oncology, Card. G. Panico Hospital of Tricase, 73039 Tricase, Italy; mtelia@libero.it (M.T.E.); emilianotamburini@yahoo.it (E.T.); 6Department of Medical Oncology, Università Vita-Salute, San Raffaele Hospital IRCCS, 20019 Milan, Italy; cascinu@yahoo.com

**Keywords:** colorectal cancer, BRAF mutation, anti-angiogenics, chemotherapy, MSI-H

## Abstract

*Background*. FOLFOXIRI plus Bevacizumab is one of the most frequently used first-line treatments for patients with *BRAF*-mutant colorectal cancer (CRC), while second-line treatment requires extensive further research. In this pooled analysis, we evaluate the impact of anti-angiogenics in patients with pre-treated *BRAF*-mutant CRC. *Methods.* We monitored patients in randomized, controlled studies who had advanced CRC and were undergoing second-line chemotherapy in addition to utilizing Bevacizumab, Ramucirumab or Aflibercept treatments. These data were pooled together with the data and results of *BRAF*-mutant patients enrolled in two phase III trials (TRIBE and TRIBE-2 study), who had been treated with second-line treatment both with or without Bevacizumab. Overall survival (OS), in relation to *BRAF* mutational status, was the primary focus. *Results*. Pooled analysis included 129 patients. Anti-angiogenics were found to have a significant advantage over the placebo in terms of OS (HR 0.50, 95%CI 0.29–0.85) (*p* = 0.01). *Conclusions*. Our pooled analysis confirms the efficacy of anti-angiogenics in pre-treated *BRAF*-mutant CRC, establishing the combination of chemotherapy plus Bevacizumab or Ramucirumab or Aflibercept as a valid treatment option.

## 1. Background

Colorectal cancer (CRC) is one of the leading causes of cancer-related deaths worldwide, with approximately half of patients developing metastases during the course of the disease [1,2]. However, owing to the considerable progress made in the treatment of metastatic disease in recent decades, median survival in modern clinical trials is around 30 months [3,4]. When defining the treatment goal in relation to metastatic disease, both patient-related (e.g., age, performance status, comorbidities) and tumor-related factors (e.g., burden of disease, number of sites involved, respectability, molecular profile, primary tumor sidedness) along with patients’ wishes should be considered. A notable driver of CRC is the mutation of the *BRAF* gene, which concerns 8–10% of colon cancers and does not overlap with *RAS* mutations. Roughly 96% of all *BRAF* mutations are a T1799A transversion in exon 15, which results in a valine amino acid substitution: V600E [5]. *BRAFV600E*-mutant CRCs (referred to in the text as *BRAF*-mutant) share the following peculiar clinico-pathological features: they are often right-sided, microsatellite-high (MSI-H), more frequent in women, have a mucinous histology and are associated with a poor prognosis [6]. However, a small fraction of *BRAF* mutations affect codons 594 and 596, representing a distinctly smaller population from a clinical point of view when compared to a *BRAFV600*-mutant. These are found more often in males, are usually left-sided, associated with *RAS* mutations, have a better prognosis and are only rarely characterized by a peritoneal relapse [7,8]. In an effort to overcome the aggressiveness of *BRAF*-mutant CRCs, some authors have proposed a more intensive regimen of chemotherapy based on the combination of 5-Fluorouracil, Leucovorin, Irinotecan, Oxaliplatin (FOLFOXIRI) and Bevacizumab [9]. The phase III TRIBE trial used randomly selected patients with metastatic CRC to receive a first-line treatment with FOLFOXIRI plus Bevacizumab versus FOLFIRI plus Bevacizumab. The trial reached its primary endpoint, which was progression-free survival (PFS), and showed significantly improved OS with a median OS of 29.8 months (95%CI 26.0–34.3) in the FOLFOXIRI plus Bevacizumab group compared with 25.8 months (95%CI 22.5–29.1) in the FOLFIRI plus Bevacizumab group (hazard ratio [HR] 0.80, 95%CI 0.65–0.98; *p* = 0.03). However, in the absence of a significant interaction between *BRAF* mutational status and the treatment arm, improved median OS was reported with the triplet subgroup that used Bevacizumab (19 vs. 10.7 months; HR: 0.84, 95%CI 0.24–1.2), albeit with a poor prognosis [9]. Although these results may be skewed by the small sample size (28 patients) in a non-preplanned analysis, FOLFOXIRI plus Bevacizumab has become a preferred option for *BRAF*-mutant CRC patients. Likewise, second-line treatment in *BRAF*-mutant CRC undoubtedly requires further research. In the overall population, the addition of anti-angiogenics Aflibercept and Ramucirumab during standard chemotherapy yielded survival improvements in their respective pivotal trials [10,11]. Furthermore, treatment with Bevacizumab beyond first-line progression demonstrated a higher survival rate in both a large observational cohort study and randomized clinical trials [12,13,14]. These data led to a major step forward in the management of pre-treated metastatic CRC patients. Subgroup analyses of VELOUR and RAISE trials suggest a potential benefit of the addition of anti-angiogenics in patients with *BRAF*-mutant CRC, although results were restricted by the small sample size [15,16]. Conversely, no data have been published about the impact of Bevacizumab on *BRAF* mutation in second-line treatment. Therefore, we performed a pooled analysis aiming at evaluating the impact of anti-angiogenics in patients with pre-treated *BRAF*-mutant CRC.

## 2. Material and Methods

### 2.1. Study Design and Inclusion Criteria

In this pooled analysis, we used randomized, controlled studies and considered patients with *BRAF*-mutant CRC treated with second-line chemotherapy plus either antiangiogenic drugs (Ramicirumab or Aflibercept) or placebo. The resulting data were then pooled with the data and outcomes of *BRAF*-mutant patients enrolled in two phase III trials (TRIBE and TRIBE-2 study). These had been treated in second-line with chemotherapy plus Bevacizumab or chemotherapy alone. A primary analysis was planned to compare OS. Ethics approval and consent to participate do not apply to this research.

### 2.2. Search Strategy

Figure 1 demonstrates the search strategy used in the meta-analysis. A bibliographic research of the PubMed, Cochrane Library, and Embase databases was conducted. Keywords used included colorectal cancer, either Aflibercept or Ramucirumab, Bevacizumab, chemotherapy, second-line, and *BRAF*. Articles published in English dating to March 2020 were retrieved. Relevant reviews and meta-analyses were also examined for the potential use of data. The annual meetings of the American Society of Clinical Oncology (ASCO and the ASCO gastrointestinal [ASCO GI] Cancers Symposium) and the European Society of Clinical Oncology (ESMO and ESMO GI) were comprehensively reviewed to detect unpublished data if pertinent.

### 2.3. Data Extraction and Management

The titles and abstracts of each of the selected studies were screened independently by two authors (A.C.G. and E.T.). The abstracts of potentially eligible trials were then read independently by the same authors who decided whether the study in question would be selected. The authors then analyzed the full text of each selected paper in order to decide the trials to be included in the pooled analysis. When there were discrepancies in trial search or selection, they discussed with a third researcher (F.G.) to reach a final consensus. The internal validity of the trial was assessed by evaluating the methods used for randomization, blindness, allocation sequence, allocation concealment and the report of missing data. All selected trials published as full-text articles in a peer-reviewed journal were analyzed and classified using the Jadad score when possible [17]. Qualitative and quantitative analyses of the selected articles were independently performed by the same two authors (A.C.G. and E.T.). When discrepancies occurred, they consulted a third researcher (F.G.) to reach a final consensus. OS were the variables under analysis.

### 2.4. Statistical Analysis

Meta-analysis was performed in accordance with the PRISMA statement recommendations [18]. Data were entered into a computer database for transfer and statistical analysis in Review Manager 5.2. Heterogeneity among the trials was assessed with a descriptive aim using the *I*2 test. *I*2 values above 50% were deemed to suggest high heterogeneity, values of 25–50% were deemed to show modest heterogeneity, and values below 25% were deemed to represent low heterogeneity. A level of 5% was assumed to be statistically significant. Differences between categorical outcome parameters were quantified using the Odds Ratio (OR) and corresponding 95%CI. Summary statistics for dichotomous outcome data were assessed using the Mantel–Haenszel method. Summary statistics for generic inverse variance data were calculated using the inverse variance method. Pooled analysis of the OR was performed using a random-effect model, assuming an error of 5% as an index of statistical significance.

## 3. Results

### 3.1. Study Selection and Characteristics

The combined search yielded 914 potentially relevant articles, 912 of which were excluded because they were either reviews, non-randomized controlled trials or had no data relating to the *BRAF*-mutant population. Two trials (RAISE and VELOUR study), with a total of 77 patients with *BRAF*-mutant CRC, were included (Figure 1).

In the RAISE trial, the authors assessed the efficacy of Ramucirumab plus FOLFIRI versus FOLFIRI alone in patients with disease progression after first-line chemotherapy. In the VELOUR trial, the authors studied the efficacy of Aflibercept plus FOLFIRI versus FOLFIRI alone in the same setting. The results of the VELOUR and RAISE trials along with their effect on patients are summarized in Table 1. Of the 1226 patients enrolled in the RAISE and VELOUR trials, 77 patients (6.3%) were *BRAF*-mutant. Both studies were deemed to be of high quality. The modified Jadad score revealed that the quality of the two individual studies was sufficient for further analysis (Table 2).

Of the 1187 patients enrolled in the TRIBE and TRIBE-2 trials, 52 patients (4.3%) were *BRAF*-mutant and received a second-line treatment. Of these, 46 (88.5%) were treated with an antiangiogenic drug and six (11.5%) with chemotherapy alone. Table 3 shows the characteristics of these patients.

### 3.2. Publication Bias and Among-Trial Heterogeneity

A total of 129 patients were included in the final pooled analysis. Figure 2 shows a funnel plot of the data. The funnel plot does not reveal significant publication bias.

No heterogeneity was found between the studies in the analysis (I^2^TEST 0%). There was a significant advantage in favor of antiangiogenics versus chemotherapy alone in terms of OS in patients with *BRAF* mutation (HR 0.50, CI95% 0.29–0.85) (*p* = 0.01) (Figure 3).

## 4. Discussion

*BRAF* mutation represents a well-recognized negative prognostic factor in patients with CRC. The worst prognoses of *BRAF*-mutant CRC have been largely discovered in both early-stage [19,20,21] and advanced-stage disease [22,23]. Many strategies have been developed to overcome the intrinsic resistance of *BRAF*-mutant CRCs, especially for those with *BRAF*V600-mutant tumors. One of these strategies is the intensification of first-line treatment by using a three-drugs regimen (FOLFOXIRI) in combination with Bevacizumab. The efficacy of this combination stems from an exploratory analysis of a phase II trial [24], subsequently echoed by the results of a subgroup analysis of the phase III TRIBE trial [9]. Preclinical evidence has shown that the MAPK pathway can increase expression of *VEGF* [25,26], thus suggesting that *RAS* and *BRAF* mutations can potentially influence the response to anti-angiogenics. Furthermore, post-hoc analyses of AVF2107g [27] and AGITG MAX trial [28] seem to suggest a numerical, although not statistically significant, survival advantage in *BRAF*-mutant CRC treated with Bevacizumab. However, besides the limited evidence resulting from retrospective analysis of small subgroups, one of the criticisms regarding the TRIBE trial concerns the added value of Bevacizumab in combination with FOLFOXIRI. In this respect, a propensity score-adjusted analysis of two randomized trials by Cremolini and colleagues demonstrated a survival advantage with the combination of FOLFOXIRI plus Bevacizumab versus FOLFOXIRI alone [29]. Despite the absence of randomized comparisons, this demonstrates the most successful attempt to answer this crucial question. Proving more difficult to treat, however, are pre-treated patients. While the introduction of *BRAF* inhibitors such as Vemurafenib, Dabrafenib and Encorafenib has recently revolutionized the treatment landscape of metastatic melanoma [30,31,32], the results in the treatment of CRC were largely unsatisfactory [33]. This can be attributed to the more complex molecular landscape of CRC compared to melanoma. Indeed, the inhibition of *BRAF* leads to a paradoxical restoration of MAPK signaling through a number of adaptive feedback mechanisms [34]. Therefore, many strategies have been developed to avoid the reactivation of the MAPK pathway and overcome the intrinsic resistance to *BRAF* inhibitors. One of these strategies is the simultaneous inhibition of a large number of components of the pathway. Recently, the results of the BEACON CRC phase III trial have been published. A total of 665 patients with pre-treated *BRAF*-mutant CRC were randomly selected to receive a triple combination of Encorafenib (a *BRAF*-inhibitor), Binimetinib (a *MEK*-inhibitor) and Cetuximab (an anti-*EGFR*), versus a double combination of Encorafenib plus Cetuximab versus an investigators’ choice (Irinotecan or FOLFIRI plus Cetuximab). Median OS was nine months for the triplet compared to 5.4 months for standard therapy (HR 0.52; *p* < 0.0001). The confirmed response rate from the blinded central review for the triplet therapy was 26% compared to 2% (*p* < 0.0001) for standard therapy [35]. Our pooled analysis seems to confirm the efficacy of anti-angiogenics in the peculiar subgroup of pre-treated patients with *BRAF*-mutant CRC. One possible explanation for the efficacy of anti-angiogenics could lie in the enrichment of this population with MSI-H tumors. In fact, in a pooled analysis of four phase III studies involving 250 *BRAF*-mutant CRCs, among deficient-MMR (dMMR) tumors, roughly one third of them had a *BRAF* mutation, while one fifth of *BRAF*-mutant also had a dMMR [36]. In this respect, in a subgroup analysis of CALGB/SWOG 80405, patients with MSI-H tumors showed longer OS in the Bevacizumab arm than in the Cetuximab arm (HR 0.13; interaction *p* < 0.001 for interaction between microsatellite status and the two arms) [37]. Although the unprecedented results of the BEACON CRC trial will hopefully change the treatment paradigm in *BRAF*-mutant CRCs, there are still patients who do not benefit from this chemotherapy-free therapy and who may potentially benefit from the combination of chemotherapy with an anti-angiogenic. Future research focusing on the biomarkers-driven selection of patients who may benefit from this triplet combination is eagerly awaited. Some limitations of this study are the limited number of trials included and the small number of patients with *BRAF*-mutant CRC enrolled in each trial. Furthermore, since the patients enrolled in each trial had different characteristics, we cannot exclude a clinical heterogeneity in our pooled analysis.

## 5. Conclusions

As of today, to the best of our knowledge, only post-hoc analyses of randomized trials have been published regarding the efficacy of anti-angiogenics in pre-treated *BRAF*-mutant CRC. Acknowledging the limitations of our pooled analysis, no definitive conclusions can be drawn and further evaluation is needed. However, recognizing that a randomized clinical trial would likely be anachronistic and unfeasible, our pooled analysis provides the best evidence available in favor of the addition of an anti-angiogenic to chemotherapy in the second-line treatment of *BRAF*-mutant CRC.

## Figures and Tables

**Figure 1 cancers-12-01022-f001:**
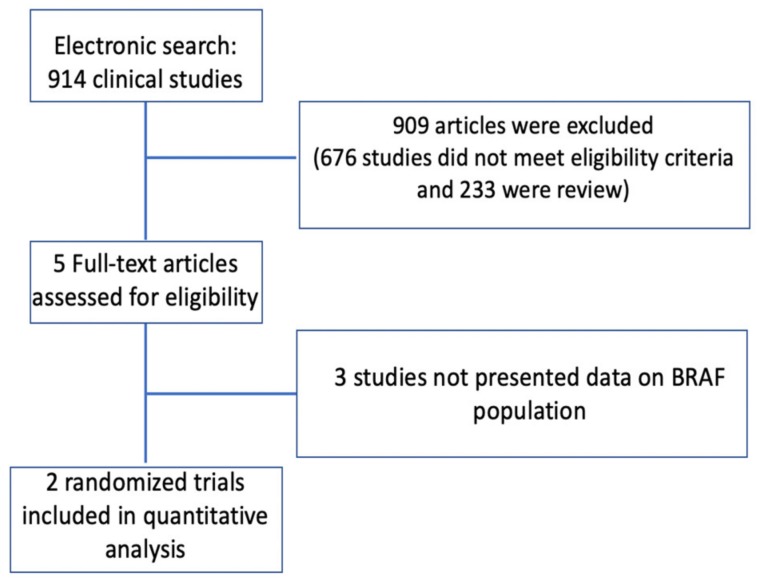
Summary of the evidence search and selection process (Flow diagram).

**Figure 2 cancers-12-01022-f002:**
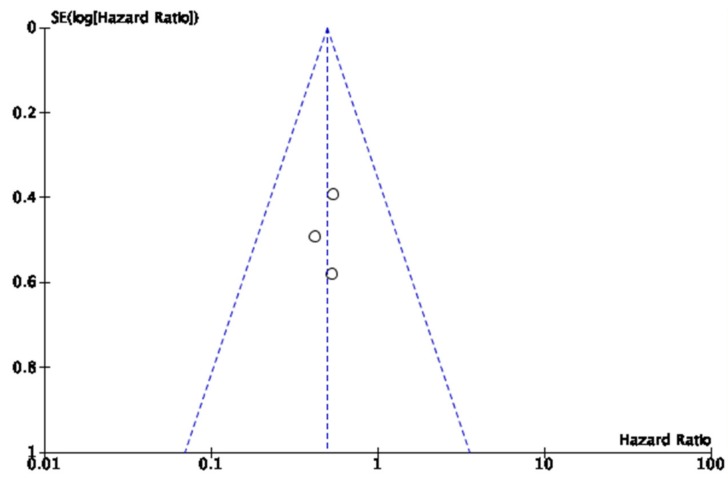
Funnel plots of publication bias.

**Figure 3 cancers-12-01022-f003:**
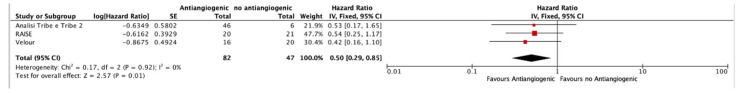
Forest plot of antiangiogenics versus no antiangiogenics in terms of OS in patients with *BRAF* mutation.

**Table 1 cancers-12-01022-t001:** Patient characteristics and results of VELOUR and RAISE studies.

	Velour	Raise Study
Experimental arm	Aflibercept plus FOLFIRI	Ramucirumab plus FOLFIRI
Prior Bevacizumab		
Yes	30.4%	100%
mOS		
Arm with antiangiogenic	13.5 months	13.3 months
Arm without antiangiogfenic	12.06 months	11.7 months
mPFS		
Arm with antiangiogenic	6.9 months	5.7 months
Arm without antiangiogenic	4.67 months	4.5 months

**Table 2 cancers-12-01022-t002:** Quality assessment of included studies using the modified Jadad score.

Raise Trial	Velour Trial	
+	+	Random sequence generation (selection bias)
+	+	Allocation concealment (selection bias)
+	+	Blinding of participants and personnel (performance bias)
+	+	Incomplete outcome data (attrition bias)
+	+	Selective reporting (reporting bias)

**Table 3 cancers-12-01022-t003:** Characteristics of *BRAF*-mutant patients in TRIBE and TRIBE-2 study.

	No. (%)
First-Line Treatment	
FOLFOXIRI + Bevacizumab	24 (46.1)
FOLFIRI + Bevacizumab	5 (9.6)
FOLFOX + Bevacizumab	23 (44.2)
Study	
TRIBE	12 (23.1)
TRIBE-2	40 (76.9)
Primary tumor location	
Right	37 (71.1)
Left	11 (21.1)
Rectum	4 (7.8)
Stage at diagnosis	
I–III	10 (19.2)
IV	42 (80.8)
Second line Therapy	
Chemotherapy plus antiangiogenic	46 (88.5)
Chemotherapy	6 (11.5)

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
