# Peer review of "The Role of Anti-Angiogenics in Pre-Treated Metastatic BRAF-Mutant Colorectal Cancer: A Pooled Analysis"

_cancers, 2020, doi:10.3390/cancers12041022_

Round 1

Reviewer 1 Report

1.Please include line and page numbering.

2.The text font and color should be consistent.

3.Have you proofread your manuscript yet? There are so many typing error.

4.In Figure.1, 909 articles were excluded. Among these 909 articles, 676 articles didn't meet critiria. What happened to the rest of it?

5.How is the methodological quality of each study?
You mentioned Jadad score in "Data extraction and management, " but there was zero information in your result section.

Author Response

1.Please include line and page numbering.

Line and page numbering have been included

2.The text font and color should be consistent.

Text font and color have been made consistent

3.Have you proofread your manuscript yet? There are so many typing error.

We proofreaded the manuscript and typing errors have been corrected

4.In Figure.1, 909 articles were excluded. Among these 909 articles, 676 articles didn't meet critiria. What happened to the rest of it?

We clarified this concept: of the 909 articles excluded, 676 were studies which did not meet the criteria and 233 were reviews

5.How is the methodological quality of each study?
You mentioned Jadad score in "Data extraction and management, " but there was zero information in your result section.

A quality assessment of included studies using the Modified Jadad score was added in the Results Section (see also Table 2)

Reviewer 2 Report

The question was well answered. This paper is thought to be ready to be published.

Author Response

The question was well answered. This paper is thought to be ready to be published.

We thank the reviewer for his comment

Reviewer 3 Report

The manuscript by Gelsomino et al. has been taken into consideration all major points proposed in the first round of revision. However, after this thorough revision I found several typos and spelling mistakes that denotes that statement in the authorship section: “FG, ACG, AS, MGV, MS, MTE, FC, CS, SC, ET, D.R., A.B., G.M., C.C., A.F. acquired data, played an important role in interpreting the results and revised the manuscript” is not true at all; which is an affront to those who are serious about authorship. Having said that, please find below my minor points:

Minor points:

-Genes goes always in italics. Please unify throughout the manuscript because I found “BRAF mutations” and “BRAF mutations”.

-Please correct: “anti-angiogenetics”, “FOLFIXIRI”…

-Abbreviations must be described at the first use: PFS or OS

-In this statement: “…improved OS was reported with the triplet plus bevacizumab

-"(19 vs 10.7 months; HR: 0.84, 95%CI 0.24-1.2) in this subgroup at poor prognosis” is mean or median?

-Fig 1. Study flowcha???

-Why authors wrote Aflibercept and Ramucirumab with capital letters and bevacizumab in lower case? Please unify.

-Latin word must be written in italics (post hoc).

Author Response

The manuscript by Gelsomino et al. has been taken into consideration all major points proposed in the first round of revision. However, after this thorough revision I found several typos and spelling mistakes that denotes that statement in the authorship section: “FG, ACG, AS, MGV, MS, MTE, FC, CS, SC, ET, D.R., A.B., G.M., C.C., A.F. acquired data, played an important role in interpreting the results and revised the manuscript” is not true at all; which is an affront to those who are serious about authorship.

We apologize for this inconvenient, typos and spelling mistakes have been carefully analyzed and corrected throughout all the text

Having said that, please find below my minor points:

Minor points:

-Genes goes always in italics. Please unify throughout the manuscript because I found “BRAF mutations” and “BRAF mutations”.

Gene names have been modified and written in Italics

-Please correct: “anti-angiogenetics”, “FOLFIXIRI”…

All these errors have been corrected

-Abbreviations must be described at the first use: PFS or OS

Abbreviations have been described at the first use

-In this statement: “…improved OS was reported with the triplet plus bevacizumab"(19 vs 10.7 months; HR: 0.84, 95%CI 0.24-1.2) in this subgroup at poor prognosis” is mean or median?

In this sentence OS is intended as median

-Fig 1. Study flowcha???

This sentence has been corrected: summary of the evidence search and selection process (Flow diagram)

-Why authors wrote Aflibercept and Ramucirumab with capital letters and bevacizumab in lower case? Please unify.

Al drug names have been modified and written with capital letter

-Latin word must be written in italics (post hoc).

Latin word has been written in italics (line 347)

Round 2

Reviewer 1 Report

  1. Of the 1187 patients enrolled in the TRIBE and TRIBE-2 trial, there is no information about how these trials were included either in Figure1 or in result section.
  2. It is probably not suitable to put a solid conclusion in your manuscript, because there were only 3 trials included in the analysis. Besides, all the included trials had small population.
  3. Although there was no statistical heterogeneity during analysis, there might be clinical  heterogeneity in the meta-analysis.  RAISE study and VELOUR study were two trials with different treatment arm. You might add further discussion about this in your manuscript.
  4. In Figure1, 2 trials were included in the analysis. However, the funnel plot and forest plot included more than 2 trials. You might want to explain these discrepancies.

Author Response

  • Of the 1187 patients enrolled in the TRIBE and TRIBE-2 trial, there is no information about how these trials were included either in Figure1 or in result section.
  • In Figure1, 2 trials were included in the analysis. However, the funnel plot and forest plot included more than 2 trials. You might want to explain these discrepancies.

REPLY point 1 and 2: In the first version of the paper with analyzed only two randomized trial. All 3 reviewers commented that we could increase the number of patients. Unfortunately, all phase 3 trial published in this setting didn’t have presented data about BRFA population, for this reason and thank of Pisa colleagues we have pooled the data of 2 randomized trials with the outcomes of BRAF-mutant patients enrolled in two phase III trials (TRIBE and TRIBE-2 study) and treated in second-line with chemotherapy plus Bevacizumab or chemotherapy alone. The data of this sub analysis had never been published previously. For this reason, in figure 1 we have inserted only two phase 3 trials (this is a flow chart of the researcher on pubmed), conversely in the funnel plot figure we have added 3 trial because final analysis was performed between 2 phase 3 trial published and the pooled analysis of TRIBE and TRIBE2 trial. For greater clarity for the reader with have added in study design and inclusion criteria that the data of TRIBE and TRIBE2 had not previously published.

  • It is probably not suitable to put a solid conclusion in your manuscript, because there were only 3 trials included in the analysis. Besides, all the included trials had small population.

REPLY: We completely agree with the reviewer and we have improved the discussion session with this point.

  • Although there was no statistical heterogeneity during analysis, there might be clinical  heterogeneity in the meta-analysis.  RAISE study and VELOUR study were two trials with different treatment arm. You might add further discussion about this in your manuscript.

REPLY: We completely agree with the reviewer and we have added in the discussion this point

Round 3

Reviewer 1 Report

The discussion can be strengthened. The manuscript is not so perfect, but can be accepted.